# The Role of Streamside Native Forests on Dissolved Organic Matter in Forested and Agricultural Watersheds in Northwestern Patagonia

**Constanza Becerra-Rodas [1,\*], Christian Little [2,3], Antonio Lara [3,4,5], Jorge Sandoval [6], Sebastián Osorio [6,7] and Jorge Nimptsch [7]**

[1] Escuela de Graduados, Facultad de Ciencias Forestales y Recursos Naturales, Universidad Austral de Chile, Valdivia 5090000, Chile

[2] Instituto Forestal (INFOR), Sede Los Ríos, Fundo Teja Norte s/n, Valdivia 5090000, Chile

[3] Center for Climate and Resilience Research (CR2, FONDAP 15110009, Universidad de Chile, Santiago 8320000, Chile

[4] Fundación Centro de los Bosques Nativos FORECOS, Valdivia 5090000, Chile

[5] Facultad de Ciencias Forestales y Recursos Naturales, Universidad Austral de Chile, Valdivia 5090000, Chile

[6] Escuela de Graduados, Facultad de Ciencias, Universidad Austral de Chile, Valdivia 5090000, Chile

[7] Laboratorio de Bioensayos y Limnología Aplicada, Facultad de Ciencias, Universidad Austral de Chile, Valdivia 5090000, Chile

[\*] Correspondence: constanza.becerra.rodas@gmail.com; Tel.: +56-9-62431088

**Abstract:** Streamside native forests are known for their key role in water provision, commonly referred to as buffers that control the input or output of nutrients from terrestrial to aquatic ecosystems (i.e., nitrogen or carbon cycle). In order to assess the functional role of indigenous forests along streamside channels, we measured 10 parameters associated with DOM (Dissolved Organic Matter) at 42 points in 12 small catchments (15–200 ha) dominated by native forests (reference, WNF), forest plantations (WFP) and agricultural lands (WAL) in which the land cover portion was calculated in the entire watershed and along 30 and 60-m wide buffer strips. We found that watersheds WFP and WAL were statistically different than WNF, according to DIC concentrations (Dissolved Inorganic Carbon) and the intensity of the maximum fluorescence of DOM components. Using linear models, we related streamside native forest coverage in buffer strips with DOM parameters. The increase of streamside native forest coverage in 60 m wide buffer strips (0–100%) was related to lower DIC concentrations (0.89 to 0.28 mg C L$^{-1}$). In watersheds WFP and WAL, the humic and fulvic-like components (0.42 to 1.42 R.U./mg C L$^{-1}$) that predominated were related to an increase in streamside native forest coverage in the form of a 60 m wide buffer strip (0–75%). This is evidence that streamside native forests influence outputs of detritus and lowered in-stream processing with concomitant downstream transport, and functional integrity and water quality. We propose that DOM quantity and quality may be a potential tool for the identification of priority areas near streams for conservation and ecological restoration in terms of recovery of water quality as an important ecosystem service. The results of this study are useful to inform policy and regulations about the width of streamside native forests as well as their characteristics and restrictions.

**Keywords:** native forests; forest plantations; agricultural lands; catchment management; dissolved organic matter; streamside native buffer; riparian vegetation

## 1. Introduction

In central Chile and northwestern Patagonia, impacts on water quality have been associated with the conversion of native forests to forest plantations (*Pinus radiata* D.Don and *Eucalyptus* spp.),

shrublands, pasturelands and agricultural lands [1–5]. These land cover changes have affected the integrity of terrestrial and aquatic ecosystems, decreasing the provision of ecosystem services such as water quality and quantity [6–8]. The best management practices associated to the maintenance of streamside native forests have been identified as key management strategies for maintaining water quality and protecting aquatic and terrestrial ecosystems [9,10]. Little and collaborators [11] observed lower dissolved inorganic nitrogen and suspended solid loads with increasing widths of streamside native forests in watersheds dominated by industrial tree plantations. Riparian vegetation has also been strongly linked to carbon biogeochemical cycles and dissolved organic matter [12,13], where inputs of allochthonous detritus from riparian forests serve as energy sources of stream ecosystems [14], especially headwater streams [15].

DOM (Dissolved Organic Matter) comprises the largest pool of transported organic matter in running waters and strongly influences river ecosystem function and nutrient cycling [16]. DOM is operationally defined as the material that passes through a filter in a range of 0.22–0.7 μm and is comprised of a mixture of organic compounds with diverse properties and ecological functions [17], depending on the origin of the organic material [18]. The quantity and quality of DOM in aquatic ecosystems can influence biological processes, such as primary production and microbial respiration, as well as chemical processes, such as photochemical reactions and heavy metal transport [17,19,20].

The decomposition of dissolved organic matter results principally in dissolved inorganic carbon (DIC) and dissolved organic carbon (DOC), these are the most common parameters for DOM quantity in natural waters [21] and major components in the carbon cycle [22]. The flux of DIC into the atmosphere from aquatic ecosystems can be significant and is enhanced by microbial respiration in streams [23]. Recently, it has been reported that concentrations of DIC have increased in some rivers with agricultural land cover, such as the Mississippi River in the United States [23,24], as well as in urbanized watersheds in the United Kingdom [25].

DOM measurements can indicate the lability of organic matter in streams, aiding in the understanding and prediction of in-stream processing versus downstream transport [20]. Fluorescence spectroscopic techniques can be used to characterize DOM quality by identifying the fluorophores that have been associated with pasture [26–28] and forest plantation land covers [22]. Thus, land cover and management practices can affect DOM characteristics and reactivity in streams [17,29]. For example, protein-like DOM fluorescence components, which are generally more labile, have been associated to watersheds containing agricultural land cover, while watersheds covered predominantly by native forests are characterized by humic-like DOM components [26,30]. Yamashita and others [22] have suggested that disturbances of forest ecosystems, such as clear cutting, affect the DOM quality in headwater streams over decades as a result of changes in the watershed´s soil organic matter characteristics due to differences in organic matter inputs.

Replacing native forests with exotic forest plantations affects the quantity and quality of DOM as well as differences in the contribution of humic-like components, as has been documented in streams located in western North Carolina, USA. Lee and Lajtha [31] observed a relatively higher proportion of protein-like DOM among harvested watersheds compared to forested (old-growth) reference watersheds in the western Cascades of Oregon. Thus, DOM quality in streams may provide key information about the terrestrial-aquatic links among ecosystems which are affected by changes in forest coverage throughout the whole watershed, as well as in the native forests in riparian areas near streams (native forest streamside buffers).

Within the headwater streams of northwestern Patagonia (Chile) and eastern Patagonia (Argentina), the primary natural source of DOM is originated by the allochthonous input from native forests [32]. There is evidence that the DOM quality in northwestern Patagonian watersheds is shifting from refractory components to more labile components due to anthropogenic sources, such as effluent from aquaculture farms [33] and agricultural practices [29], which is being discharged into streams and rivers. Cuevas et al. [9,34], and Little et al. [11] have documented exports of nutrients in watersheds dominated by different vegetation coverages and streamside buffer. However, to the best of our

knowledge there are no detailed studies relating DOM quantity and quality with the vegetation coverage of an entire watershed, as well as with streamside native forest coverage in buffer strips located in watersheds dominated by forest plantations (WFP) or agricultural lands (WAL). We used reference watersheds (WNF) dominated by Valdivian temperate rainforest. We hypothesized that DOM quantity and quality in watersheds dominated by forest plantations and agricultural lands would be regulated by the presence of streamside native forests in buffer strips. Specifically, we hypothesized that an increase in streamside native forest coverage would be accompanied by more refractory DOM components in streams, with added structural complexity and more downstream transport. We aimed to document how DOM quantity and quality can serve as useful indicators of water quality that can be related to land use/land cover at different scales, such as the width of the streamside native forests. This knowledge is useful to inform decision-making regarding the conservation, management, and ecological restoration in northwestern Patagonian watersheds.

## 2. Materials and Methods

### 2.1. Study Area

The study area corresponded to a set of 12 small watersheds located in the Iñaque River Basin, northwestern Patagonia (39.6° S). This area is characterized by intensive agricultural, livestock and forestry activities, the last of which is associated with *Pinus radiata* D. Don and *Eucalyptus* spp. plantations for the pulp and lumber industries (Figure 1). The climate of the study area is classified as oceanic wet temperate with a Mediterranean influence (Cfsc) [35], characterized by 1800 mm of annual precipitation concentrated between April and August with relatively dry summers (January–March).

We sampled 42 sites in stream reaches in watersheds dominated by native forests (reference condition, WNF), forest plantations (WFP) and agricultural lands (WAL), Figure 1 and Table 1. These watersheds were selected based on the dominant land cover class determined in accordance with the National Vegetation GIS Map from Corporación Nacional Forestal (CONAF) [36]; they were also accessible by road and had similar topography and soil characteristics.

**Table 1.** Characteristics of watersheds dominated by native forests (WNF), forest plantations (WFP) and agricultural lands (WAL) according to sampled sites, elevation range, number of sampled locations of DOM (Dissolved Organic Matter) and description of the land cover.

| Watersheds | Sampled Sites | Drainage Area (ha) | Elevation Range (m a.s.l.) | Number of Locations Sampled for DOM Quantity | Number of Locations Sampled for DOM Quality | Description of Land Cover in Entire Watershed | Description of Land Cover in 30 and 60-m Wide Buffer Strips |
|---|---|---|---|---|---|---|---|
| WNF | A | 144 | 270–384 | 5 | 20 | Old-growth forest/Mixed broadleaved evergreen (*Laureliopsis philippiana*) | Second-growth forest/Mixed broadleaved evergreen (*Laureliopsis philippiana, Chusquea quila* and *Aristotelia chilensis*) |
| | B | 29.6 | 246–367 | 5 | 20 | Second-growth forest/Mixed broadleaved evergreen. (*Laureliopsis philippiana* and *Nothofagus obliqua*) | |
| | C | 26.1 | 233–246 | 2 | 8 | | |
| | D | 17.3 | 228–232 | 2 | 8 | | |
| WFP | A | 42.8 | 98–153 | 3 | 12 | Industrial plantation of *Eucalyptus nitens* and *Eucalyptus globulus* (14 years) | Second-growth forest/Deciduous (*Nothofagus obliqua*) |
| | B | 66.6 | 82–172 | 3 | 12 | Industrial plantation of *Eucalyptus globulus* (14 years) and Plantation of *Pinus radiata* (3 years after clear-cutting). | Second-growth forest/Mixed broadleaved evergreen (*Laureliopsis philippiana, Chusquea quila* and *Aristotelia chilensis*) |
| | C | 15.4 | 97–180 | 4 | 16 | Industrial plantation of *Pinus radiata* (3 years after clear-cutting). | |
| | D | 79.1 | 96–139 | 4 | 16 | Industrial plantation of *Pinus radiata* (14 years) | |

<div align="center">**Table 1.** *Cont.*</div>

| Watersheds | Sampled Sites | Drainage Area (ha) | Elevation Range (m a.s.l.) | Number of Locations Sampled for DOM Quantity | Number of Locations Sampled for DOM Quality | Description of Land Cover in Entire Watershed | Description of Land Cover in 30 and 60-m Wide Buffer Strips |
|---|---|---|---|---|---|---|---|
| WAL | A | 145 | 246–274 | 3 | 12 | Grasslands of *Holcus lanatus* | Second-growth forest/Deciduous (*Nothofagus obliqua*, *Chusquea quila*) |
| | B | 108 | 169–245 | 4 | 16 | | |
| | C | 21 | 201–284 | 3 | 12 | | |
| | D | 219.5 | 279–295 | 4 | 16 | Grasslands of *Holcus lanatus* and presence grazing animals | |
| TOTAL | 12 | | | 42 | 168 | | |

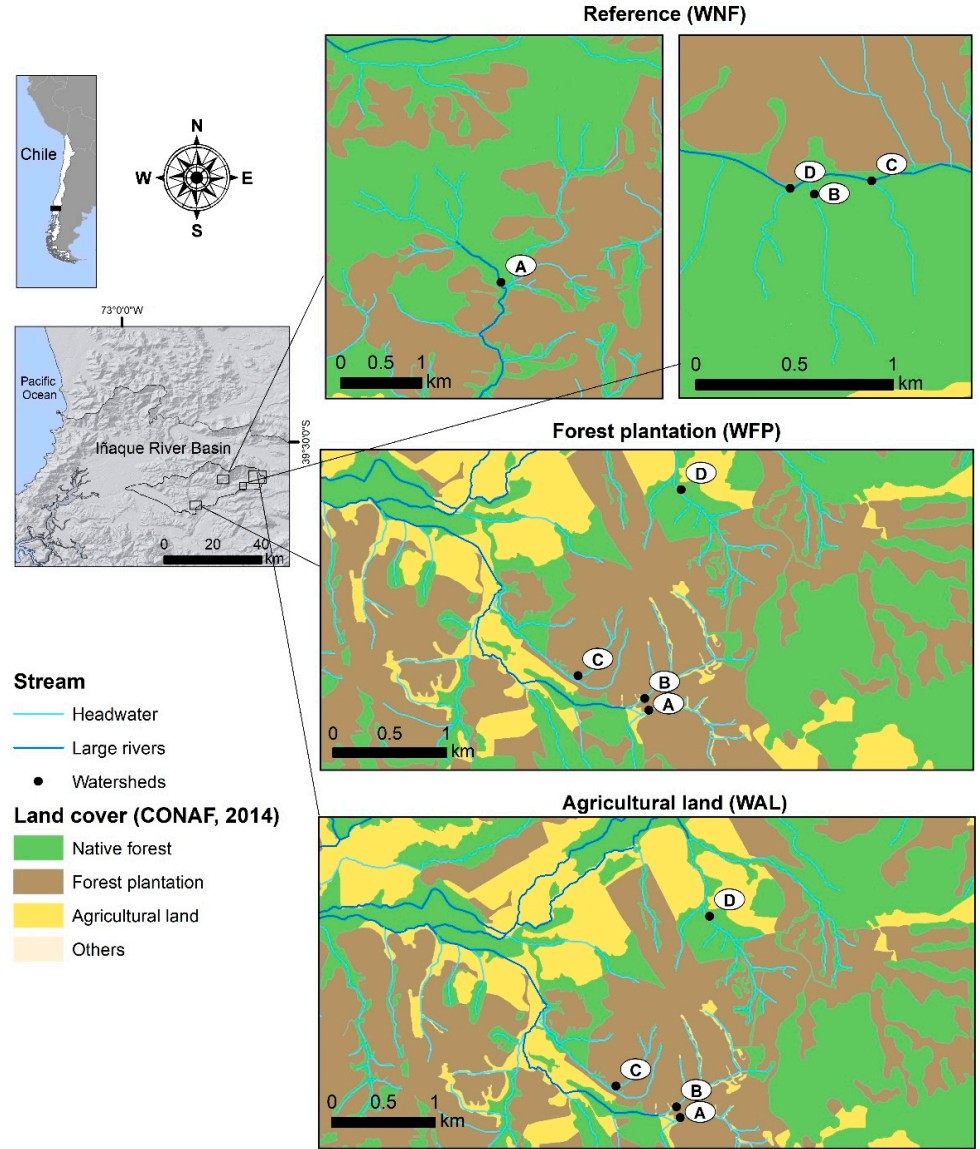

**Figure 1.** Map of the 12 sampled sites in the Iñaque River Basin. Letters represent watersheds dominated by native forests (WNF), forest plantations (WFP) and agricultural lands (WAL).

## 2.2. Spatial Analysis

The classification of land cover followed the protocol described by Lara and Sandoval [37]. This included the following classes: 1) native forest (NF), 2) mature exotic forest plantation (MFP), 3) young exotic forest plantation (YFP) (i.e., <5 years), and 4) agricultural land (A). For each sample location, watersheds and buffer strips were delineated with spatial analysis using DEM data. A buffer

tool was used to define buffer strips of either 30 or 60-m wide on each side of the stream (Figure 2).
Areas of each of these four land cover classes were determined for the entire watershed at a 1:50,000
scale provided by the Corporación Nacional Forestal (CONAF) [36] and buffer strips (30 and 60-m
wide) were delineated by manual photointerpretation based on color satellite imagery at a 1:5000
scale freely provided by Google Maps [38] and verified with ground truthing techniques [39]. Final
maps were developed and incorporated into a geographic information system using Quantum GIS
(QGIS) [40].

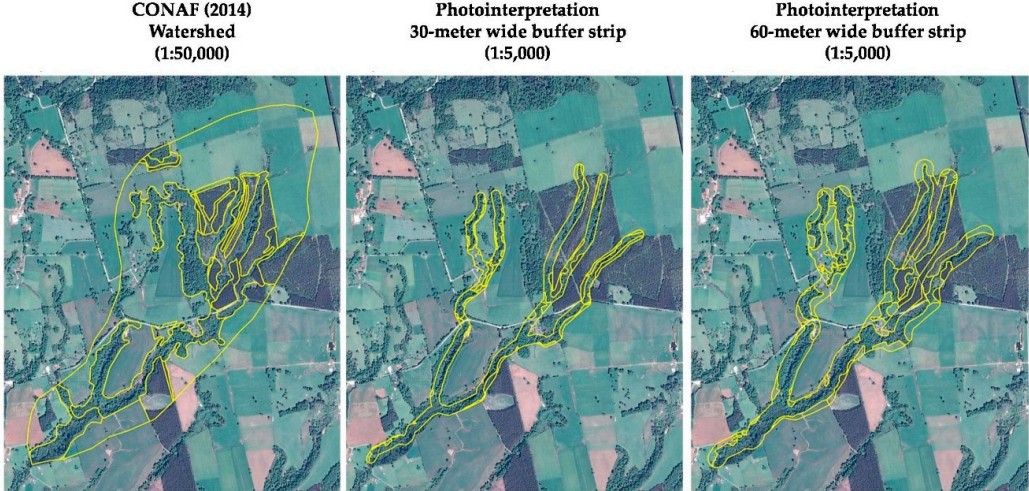

**Figure 2.** Example of the mapping and assessment of land use/land cover classes in the entire watershed
and 30 and 60-m wide buffer strip with through the photointerpretation of color satellite imagery
viewed in ©2015 Google Maps [38]; buffers were delimited with QGIS [40].

Streamside native forests were expressed as the percentage of native forest (NF) cover found
in 30 and 60-m wide buffer strips, which ranged from 32% to 78% and 18% to 80% in watersheds
dominated by forest plantations (WFP, Figure 3a) and agricultural lands (WAL, Figure 3b), respectively.

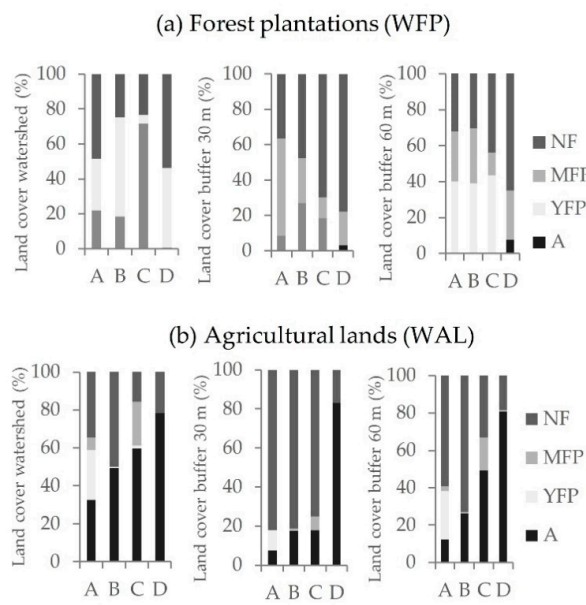

**Figure 3.** Land cover of NF: Native forest; MFP: Mature forest plantation; YFP: Young forest plantation;
and A: Agricultural land; in the entire watershed, and in 30 and 60-m wide buffer strips for watersheds
dominated by (**a**) forest plantations (WFP) and (**b**) agricultural lands (WAL).

## 2.3. Data Collection and Laboraroty Analysis

Water samples were collected from surface streams throughout each watershed in the austral winter (July and August) during winter baseflow conditions, when it is likely that soils were near saturation and the stream water reflected the terrestrial chemistry signal [31]. Discharge streamflow curves were recorded by Dirección General de Aguas (DGA) (Figure S1). To determine DOM quantity and quality, water samples were collected in *situ* using carbon free borosilicate amber vials, filtered through 0.22 μm Millex—GP Hydrophilic PES filters into acid washed borosilicate amber glass containers (Chem-Merck), and acidified to pH 2 with 100 μL of concentrated HCl (Merck). For transport to the laboratory, all water samples were immediately cooled with ice water and stored at max. 4 °C. DOM quantity and quality measurements were performed within 48 h of sampling.

DOM quantity was measured as DOC concentrations using a high temperature catalytic combustion carbon analyzer (High TOC-ELEMANTAR). The DOC concentration was estimated by subtracting the DIC concentration from the total dissolved carbon concentration [33].

DOM quality was assessed using a Cary Eclipse spectrofluorometer for fluorometric analysis. For each sample, excitation-emission matrices (EEMs) standardized to Raman units were generated. The identification and validation of fluorophores was completed using a parallel factor analysis (PARAFAC) and subsequent split-half validation. Each resulting PARAFAC component (Table 2, Figure S2) represented a group of fluorophores with specific fluorescence characteristics [41,42]. These components were compared to the values of components described in the OpenFlour database [43], indicating that components C1 and C2 were related to terrestrial material, since they were comprised of humic-like fluorophores. Component C3 was similar to tryptophan-like components (i.e., amino acids which are present in low concentrations in natural waters [44]) and component C4 was characterized by protein-like characteristics. This analysis was carried out in MATLAB version 2009 [45], according to Stedmon and Bro [46]. The intensity of the maximum fluorescence of each fluorophore was represented by Fmax (R.U., Raman Units) [47].

We calculated the $SUVA_{254}$ index, a proxy for DOM aromaticity, utilizing the absorbance spectra [48]. We calculated the following indexes from the fluorescence EEMs: (a) a humification index (HIX) [49], where higher values indicate an increasing humification and values near 1 or 2 indicate non-humified plant material; (b) a fluorescence index (FI) [19], where values near 1.8 suggest predominant microbial sources (i.e., autochthonous or microbially altered terrestrial C, and values less than 1.3 suggest terrestrial material derived from allochthonous sources; and (c) a freshness index (β:α) [50,51], where values over 1 indicate that DOM is autochthonous and values less than 0.6 indicate that DOM is allochthonous.

**Table 2.** Maximum excitation and emission wavelengths of four components identified by the parallel factor analysis (PARAFAC) model and their sources.

| Components | Excitation Max (nm) | Emission Max (nm) | Name Component | Sources |
|---|---|---|---|---|
| Component 1 (C1) | 240 | 418.5 | Similar to humic-like | Terrestrial material |
| Component 2 (C2) | 240 | 486.5 | Similar to fulvic-like | Terrestrial material |
| Component 3 (C3) | 280 | 32.6 | Similar to tryptophan-like | Proteins or less degraded peptide material |
| Component 4 (C4) | 240 | 338 | Similar to protein-like | Autochthonous or microbially altered terrestrial |

## 2.4. Data Analysis

In order to assess significant differences between different conditions compared with the reference, we used non-parametric analysis of variance, Kruskal-Wallis (ggpubr R packages [52]) and Nemenyi tests of multiple comparisons for independent samples (Tukey) (PMCMR R package [53,54]).

These analyses identified a subset of DOM parameters which differed significantly for each watershed type ($p < 0.05$, Table 3 and Table S2). Spearman rank correlations (cor.test function in R corrplot R package [55]) related DIC concentrations and intensities of the maximum fluorescence of PARAFAC components to the land cover in eight watersheds (WFP and WAL). Land cover included the percent of land covered by native forest (NF), mature forest plantation (MFP), young forest plantation (YFP) and agricultural land (AL) as a proportion of the entire watershed and 30 and 60 m wide buffer strips. Finally, the two DOM parameters that differed significantly for WFP and WAL (DIC concentrations and sum of C1 and C2 components, and sum C3 and C4 components) were related to the amount of streamside native forest found in buffer strips according to a linear model (lm function in R). Statistics were performed using R version 3.2 [56], Rstudio [57], ggplot, R package designed to create and customize plots [58].

**Table 3.** Average (±standard deviation) values of Dissolved Organic Carbon (DOC) and Dissolved Inorganic Carbon (DIC) concentrations, indexes and PARAFAC components (C1, C2, C3 and C4). The intensities of fluorescence of each PARAFAC component were standardized by the DOC concentration of each sample. The statistical differences among watersheds dominated by native forests (WNF), forest plantations (WFP) and agricultural lands (WAL) are shown in superscript letters.

| Title | WNF | WFP | WAL |
|---|---|---|---|
| DIC (mg C L$^{-1}$) | $0.16 \pm 0.09^A$ | $0.52 \pm 0.27^B$ | $1.07 \pm 0.5^C$ |
| DOC (mg C L$^{-1}$) | $0.16 \pm 0.06$ | $0.17 \pm 0.09$ | $0.15 \pm 0.04$ |
| FI | $1.45 \pm 0.8$ | $1.82 \pm 1.29$ | $1.45 \pm 0.41$ |
| HIX | $8.9 \pm 10.1^A$ | $3.07 \pm 2.78^B$ | $3.94 \pm 1.72^A$ |
| β: α | $0.35 \pm 0.21^A$ | $0.51 \pm 0.17^B$ | $0.41 \pm 0.21^{AB}$ |
| SUVA$_{254}$ (L mg$^{-1}$ m$^{-1}$) | $4.48 \pm 2.51$ | $3.99 \pm 2.29$ | $6.01 \pm 8.45$ |
| C1 (R.U./mg C L$^{-1}$) | $0.34 \pm 0.13^A$ | $0.42 \pm 0.18^A$ | $0.61 \pm 0.19^B$ |
| C2 (R.U./mg C L$^{-1}$) | $0.28 \pm 0.12^A$ | $0.35 \pm 0.17^{AB}$ | $0.38 \pm 0.11^B$ |
| C3 (R.U./mg C l$^{-1}$) | $0.08 \pm 0.09^A$ | $0.19 \pm 0.13^B$ | $0.22 \pm 0.19^B$ |
| C4 (R.U./mg C l$^{-1}$) | $0.20 \pm 0.22^A$ | $0.42 \pm 0.30^B$ | $0.30 \pm 0.29^B$ |

## 3. Results

### 3.1. Dissolved Organic Matter (DOM) Quantity and Quality

Watersheds dominated by native forests, forest plantations and agricultural lands, presented different average DIC concentration values of 0.16 mg L$^{-1}$ (WNF), 0.52 mg L$^{-1}$ (WFP) and 1.07 mg L$^{-1}$ (WAL), respectively (the *p*-value was less than 0.05, Table 3). When comparing watersheds dominated by forest plantations (WFP) and agricultural lands (WAL) with the reference condition (WNF), we found no significant differences in average values of DOC concentrations, the fluorescence index (FI), or the SUVA$_{254}$ concentration ($p > 0.05$, Table 3, Table S2). The humification index (HIX) in watersheds dominated by native forests (WNF) and agricultural lands (WAL) presented average values of 8.94 and 3.94, respectively (Table 3), significantly higher than those found in watersheds dominated by forest plantations (WFP), which averaged HIX values near 3.07. The intensity of the maximum fluorescence of component C1 in watersheds dominated by native forests (WNF) and forest plantations (WFP) was 0.34 and 0.42 R.U./mg C L$^{-1}$, respectively (Table 3). These average values were significantly less than those in watersheds dominated by agricultural lands (WAL), which presented values close to 0.61 R.U./mg C L$^{-1}$ ($p < 0.05$). The intensity of the maximum fluorescence of component C2 in reference watersheds (WNF) presented values near 0.28 R.U./mg C L$^{-1}$ and were significantly lower than watersheds dominated by agricultural lands (WAL), with values near 0.38 R.U./mg C L$^{-1}$ (Table 3). The intensity of the maximum fluorescence of tryptophan-like components (C3) within reference watersheds (WNF) presented average values of 0.08 R.U./mg C L$^{-1}$, while in watersheds dominated by forest plantations (WFP) and agricultural lands (WAL), values were close to 0.19 and 0.22 R.U./mg C L$^{-1}$, respectively (Table 3). Protein-like components (C4) in reference watersheds (WNF)

presented intensity of the maximum fluorescence values near 0.2 R.U./mg C L$^{-1}$, significantly lower than those in watersheds dominated by forest plantations (WFP, 0.42 R.U./mg C L$^{-1}$) and agricultural lands (WAL, 0.3 R.U./mg C L$^{-1}$), $p < 0.05$, Table 3).

### 3.2. Relating Dissolved Organic Matter (DOM) Quantity and Quality and Streamside Native Forests

Streamside native forest coverage was determined as the percentage of land covered by native forests in 30 and 60 m wide buffer strips (Figure S3). DOM quantity and quality were thus associated with streamside native forest coverage. In watersheds dominated by forest plantations (WFP), DIC concentrations were negatively related to streamside native forest coverage in 60 m wide buffer strips ($R^2 = 0.43$, $p < 0.05$, Figure 4a). In watersheds dominated by agricultural land (WAL), the intensity of the maximum fluorescence of humic-like components (C1 and C2), standardized by DOC concentrations, was positively related to streamside native forest cover in 60 m wide buffer strips ($R^2 = 0.12$, $p < 0.05$, Figure 4b).

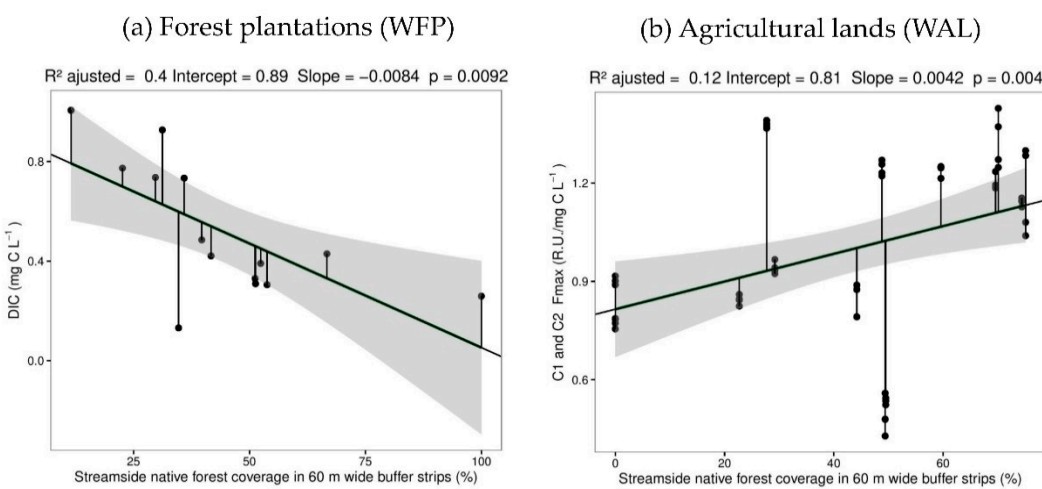

**Figure 4.** Relating DOM (Dissolved Organic Carbon) quantity and quality with streamside native forests. (**a**) DIC concentration in watersheds dominated by forest plantations (WFP) with 60 m wide streamside native forest ($n = 14$); (**b**) The intensity of fluorescence max. of components C1 and C2 in watersheds dominated by agricultural land (WAL) with streamside native forests coverage in buffer strips of 60 m wide ($n = 56$). Grey color indicates 98% confidence region.

## 4. Discussion

Assessing areas of land cover classes at different scales (i.e., in the entire watershed and in 30 and 60 m wide buffer strips, Figure S3), our study demonstrated that streamside native forest cover influences DOM (Dissolved Organic Carbon) quantity and quality in watersheds dominated by forest plantations (WFP) and agricultural lands (WAL). This outcome is possible due to the studied watersheds' differences in land cover in their buffer strips, as well as significant differences in the DOM quantity and quality found in runoff from watersheds dominated by forest plantations and agricultural lands, in contrast to the reference condition [27]. Watersheds dominated by forest plantations included a gradient of the percent of streamside native forest coverage found in their buffer strips, which controlled DIC (Dissolved Inorganic Carbon) concentrations. Increased DIC concentrations may be related to changes in allochthonous inputs of more labile DOM, which is more readily degraded, thus increasing microbial respiration and the mineralization of carbon [23]. Increased microbial respiration and carbon mineralization could be associated with diffuse sources from fertilizers, herbicides, and cattle manure, which can contribute to the eutrophication of streams [59]. DIC concentrations reported in this study range from 0.05 to 1.29 mg C L$^{-1}$ which are less than the average value reported by Raymond et al. [24] for tributaries of the Mississippi River (mean values of 6 to 40 mg C L$^{-1}$, measure of alkalinity). Higher DIC concentrations reflect fast biological processing of DOM in headwater

streams [60]. DOC (Dissolved Organic Carbon) concentrations found in this study were not significantly different ($p > 0.05$) in watersheds dominated by native forests, forest plantations and agricultural lands. This could be explained by the fact that DOC concentrations are primarily controlled by hydrological and climatic factors, as described by other authors [22,26,27,34]. DOC concentrations from watersheds dominated by Valdivian temperate rainforests reported in this study (0.1 to 0.21 mg C $L^{-1}$) are similar to the average values reported by Nimptsch et al. [33] in control samples taken from the Molco river associated with Nothofagus-dominated temperate rainforests (0.1 to 0.3 mg C $L^{-1}$). These values are less than the average values reported by Yamashita et al. [22] for rivers in western North Carolina, USA (1.8 mg C $L^{-1}$), Graeber et al. [26] for rivers in northern Germany (1.3 to 3.8 mg C $L^{-1}$) and Lajtha and Jones [61] for rivers in western Oregon, USA (1.2 to 2 mg C $L^{-1}$), where watersheds were dominated by mixed and coniferous forests.

In this study, the core consistency of the four PARAFAC components, as well as the lability of fluorophores as described by Fellman et al. [62], allowed for the characterization of DOM quality at its origin. The intensity of the maximum fluorescence of the four components observed in watersheds dominated by native forests were significantly lower than watersheds dominated by forest plantations and agricultural lands. The freshness index in watersheds dominated by native forests were significantly lower than watersheds dominated by forest plantations. Some studies have indicated that humic components were derived primarily from higher plants [47,62,63], for example, from terrestrial sources such as surface runoff during storm events in Lee et al. [27]. In this study, watersheds dominated by agricultural lands presented a higher intensity of the maximum fluorescence, mainly of humic-like and fulvic components (C1 and C2), commonly found in all types of environments [47]. In watersheds dominated by agricultural lands, the intensity of the maximum fluorescence of protein-like components was higher, which may be due to allochthonous contributions consisting of carbohydrates, lipids and proteins that are re-mineralized or assimilated by bacterial action or autochthonous production [19,30,63]. This lability of DOM can affect the respiration rate in streams, impacting biota and $CO_2$ exchange with the atmosphere. Previous studies have found that watersheds dominated by agricultural lands have presented more protein-like components produced by aquatic microorganisms in streams. These watersheds also proved to have the highest freshness index (β:α) or the greatest amount of autochthonous DOM [50,64]. The degree of humification (HIX) and freshness index (β:α) in watersheds dominated by agricultural land indicate that primarily humic-like components dominant from terrestrial sources.

Watersheds dominated by forest plantations showed a high intensity of the maximum fluorescence of humic-like and protein-like components. Yamashita et al. [22] reported increased protein-like components in watersheds dominated by forest plantations using clear-cutting practices. According to Lee and Lajtha [31] harvested watersheds had less input of coarse woody debris than reference watersheds. However, since $SUVA_{254}$ did not differ among watersheds with different land covers it was not possible to determine the refractory or labile composition of DOM [41].

The characteristics of DOM within watersheds dominated by forest plantations and agricultural lands may be partially attributed to the amount of streamside native forest coverage. Our study indicates that 60-m wide buffer strips, which is more than required by Chilean law for headwater streams [65], effectively influence DOM quantity and quality, specifically DIC and humic and fulvic-type components. Watershed A was dominated by forest plantations (WFP), specifically a young forest plantation planted 3 years after clear-cutting. This plantation reached the buffer strip all the way to the edge of the stream with no native forest buffer and had high levels of PARAFAC component C3, similar to the tryptophan-like component, possibly due to changes in litter inputs and more solar radiation, which could have affected DOM photodegradation [22].

We developed a conceptual model of the key role that streamside native forests play in controlling DOM quantity and quality in watersheds dominated by anthropogenic activities (forest plantations and agricultural lands, Figure 5). We hypothesize that the streamside native forest coverage of headwater streams (i.e., streamside buffer strips of 60-m) influences outputs of allochthonous sources, with

more refractory DOM dominating from terrestrial sources (i.e., soil and plant leachates), with lower bioavailability downstream.

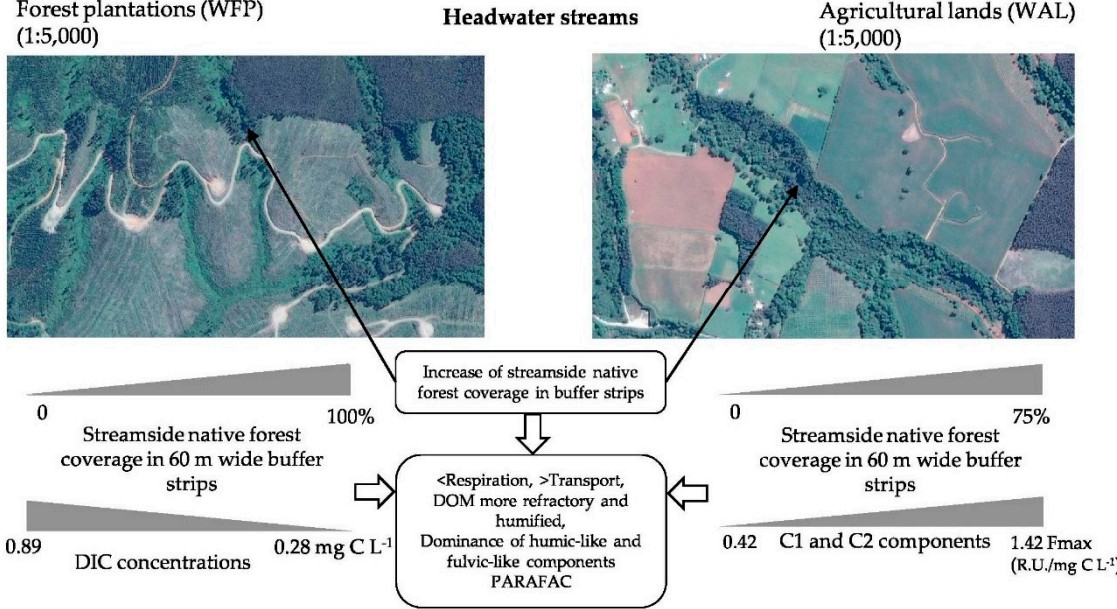

**Figure 5.** Conceptual model of the role of native forests in buffer strips and variations of the characteristics of Dissolved Organic Matter (DOM) in headwater streams dominated by forest plantations and agricultural lands with an increase in streamside native forest coverage.

The conversion of native forests to agricultural lands and forest plantations in buffer strips has consequences for terrestrial and river ecosystems downstream, impacting water quality for biota and human use [59,66]. It should be noted that samples in this study were collected only in the winter, and seasonal variations of DOM should be assessed.

## 5. Conclusions

This study provides new insights into the role of streamside native forests in influencing outputs of allochthonous sources and refractory DOM within watersheds impacted by anthropogenic activities in northwestern Patagonia, Chile. These findings provide baseline information on how the amount of streamside native forests may affect carbon pools in streams and rivers. Our study suggests that streamside buffer strips of 60-m wide dominated by native forests be considered as a standard to protect water provision. The role of streamside native forests in regulating water quality should be considered when evaluating catchment management strategies and carbon budgets of terrestrial-fluvial-atmospheric systems. We suggest that DOM quantity and quality document the role of streamside native forests in influencing ecological functions in terrestrial and aquatic systems. Moreover, rapid assessment of watershed DOM may be used as a tool for identifying priority sites for conservation and restoration of native riparian forests essential for the recovery of ecosystem services in northwestern Patagonian headwater streams. The results of this study are useful to inform policy and regulations about the width of native forest buffers as well as their characteristics and restrictions.

**Supplementary Materials:** The following are available online at http://www.mdpi.com/1999-4907/10/7/595/s1, Figure S1: Streamflow recorded in the Iñaque River Basin gage, for the hydrologic year in which the samples were taken. Black squares represent sampling dates (23 July, 4, 7, 10, and 14 August 2015), Figure S2: EEMs dissolved organic matter fluorescence component results from parallel factor analysis (PARAFAC) split-half validated (100 iterations, 4 component model), Figure S3. Spearman correlation matrices for watersheds dominated by forest plantations and agricultural lands according to the quantity and quality of DOM (DIC, Humic and Protein) and four types of land cover (nf: native forest; mpt: mature forest plantation; young forest plantation; and agr: agricultural land in the entire watershed (w), streamside native forest coverage in 30 m (b30) and 60 m (b60) wide

buffer strips. Intensity of the color of the circle indicates degree of correlation (red indicates a negative relationship and blue indicates a positive relationship); Size of the circles indicates statistical significance ($p < 0.05$); White squares indicate that there is no statistical significance ($p > 0.05$), Table S1. Areas of four land cover classes: native forest (NF), mature forest plantation (MFP), young forest plantation (YFP) and agricultural land (A) in the entire watershed and buffer strips of 30 and 60-m widths in watersheds dominated by native forests or reference (WNF), forest plantations (WFP) and agricultural lands (WAL), Table S2: Nemenyi Test of multiple comparisons.

**Author Contributions:** C.B.-R., C.L. and J.N. designed the study, C.B.-R., J.S. and S.O. conducted monitoring campaigns, C.B.-R., C.L., A.L., and J.N. analyzed the data, C.B.-R., C.L., J.S., and S.O. writing—original draft preparation, C.B.-R., C.L., A.L., and J.N. writing—review.

**Funding:** C.B.-R.: C.L. and A.L. supported by Center for Climate and Resilience Research (CR2) (CONICYT/ FONDAP/15110009).

**Acknowledgments:** The authors acknowledge the valuable support of the staff from Laboratorio de Bioensayos y Limnología Aplicada, UACh, Valdivia, Chile, A.L. acknowledges the support of CONICYT PAI-MEC 80170046, C.BR. and A.L. acknowledge the support of project 020/2016 – VII Concurso del Fondo de Investigación del Bosque Nativo, Fundación Centro de los Bosques Nativos FORECOS, Valdivia, Chile. J.N. acknowledge the support of Kevin Ryan and Escuela de Graduados, Magíster en Recursos Hídricos, Facultad de Ciencias, Universidad Austral de Chile, Valdivia, Chile.

**Conflicts of Interest:** The authors declare no conflict of interest.

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
