# Peer review of "The Role of Streamside Native Forests on Dissolved Organic Matter in Forested and Agricultural Watersheds in Northwestern Patagonia"

_forests, doi:10.3390/f10070595_

Round 1

Reviewer 1 Report

The manuscript “The role of native riparian forests on dissolved organic matter in forest plantation and agriculture watersheds in northwestern Patagonia” investigates the influence of four different riparian forest types on the water quality, indicated by several measures related to dissolved organic matter. The manuscript has a the potential for being a significant contribution to gain knowledge about the relationship of different riparian buffers to water quality. 

So far the methods are weakly described and results and conclusions lack clarity. It is not clear what the conclusions are based on. Clarifying methods, results and improving English could lead to a significant contribution, but at the current state, the manuscript is not fulfilling the minimum standards for publication.

Title: What is an ‘agriculture watershed’? This seems to be an uncommon shortcut to ‘agriculturally dominated watersheds’. 

Abstract:

In the abstract, the authors talk about three land cover types, but in the analysis four land cover types (young and old plantation) are distinguished. Consistency in communication avoids confusing the reader.

L30: ‘60 and 30 m width buffer’. It is unclear what is meant. I suppose the authors meant ‘the difference between 30 and 60 m wide buffers’.

L30: linear models instead of lineal

L34: What is meant by ‘dominance input’

L33-35: Unclear what the result is: ‘In watersheds dominated by forest plantations, lower DIC is correlated with higher native forest cover.’ Why distinguishing watersheds into three classes according to main forest cover type and then further correlating native forest cover with DIC?

L37: Unclear what is meant by ‘DOM metrics’

Methods

L177: Why would you perform a Shapiro-Wilk test for land cover data and DOM metrics? You are hopefully aware that the assumption of an ANOVA and a linear model are the normality, which means the normal distribution of theresiduals, and the homogeneity of variance.

L184: What are the DOM quantity and quality parameters? Be consistent in the terminology used! DOM metrics, DOM characteristics, DOM parameters. Are these all the same? And what are the qualitative ones? What are ‘the land cover data’ - provide the variables used in the analysis. And what are the ‘scales of watersheds’?

L185: Why the z-standardization?! You use rank based approaches. Z-Standardization doesn’t make sense here.

Figure 1: How are the three maps with A to L related to each other? Provide a map of the basin where map sections are related to each other. It looks like a very nested design, where all native forest areas are closer to each other than to the other forest types.

L192: Specify the two independent variables. ‘Land cover of native forests to scales of analysis riparian area at 60 and 30 m.’ This is not a proper English sentence, and it is unclear what exactly the (2, 3 or more?) independent variables are.

Methods are unclear. As far as I understand you use ANOVA plus PostHoc test with the Null-Hypothesis: ‘There are no significant differences in the DOM between the four land cover types: native forest, young plantation, old plantation, agriculture in the watersheds A to L’. But apparently DOM is measured by several indicators, so you have several ANOVAs. How many? Consider multiple testing!

The second step is the Spearman correlation of DOM parameters (still unclear of how many and which ones they are) and the land cover data. What variables were correlated? Consider again multiple testing! Why (according to Figure S3) the Spearman correlation was performed only for the forest plantation and agriculture watershed?

The third step are several linear models, as far as I understand with the same response variables than in the ANOVA, only that there are several more independent variables. Then the linear models are redundant to the ANOVA and it is unclear why to use both.

Results

L205: Replace 0,05 by 0.05 and be consistent in the use of a digit separator.

L207: Freshness index. It is the first time you talk about such thing and it is completely unclear where it comes from, or whether it is any synonym to something you have talked about before. If this is something measured or calculated it needs to be mentioned in the methods.

Conclusion

L319 New insights – is unspecific. Please specify what are these insights.

Author Response

Response to Reviewer 1 Comments

Title: What is an ‘agriculture watershed’? This seems to be an uncommon shortcut to ‘agriculturally dominated watersheds’.

Response 1:  we modificated to “The role of native riparian forests on dissolved organic matter in watersheds dominated by forest plantations and agricultural lands, northwestern Patagonia”

Abstract:

In the abstract, the authors talk about three land cover types, but in the analysis four land cover types (young and old plantation) are distinguished. Consistency in communication avoids confusing the reader.

Response 2:  Are three type of watershed dominated by native forests (reference watersheds), forest plantations, and agricultural lands.

Relative area for four types land cover as independent variables (native forest (NF), mature forest plantation (MFP), young forest plantation (YFP) and agricultural land (A)) in the entire watershed and riparian areas (60 and 30 meter wide buffers).

DOM quantity and quality as dependent variables.

L30: ‘60 and 30 m width buffer’. It is unclear what is meant. I suppose the authors meant ‘the difference between 30 and 60 m wide buffers’.

Response 3:   changed to wide buffers

L30: linear models instead of lineal

Response 4:   changed to lineal

L34: What is meant by ‘dominance input’

Response 5:  we changed redaction highest proportions of DOM more refractory

L33-35: Unclear what the result is: ‘In watersheds dominated by forest plantations, lower DIC is correlated with higher native forest cover.’ Why distinguishing watersheds into three classes according to main forest cover type and then further correlating native forest cover with DIC?

Response 6: Because we grouped watersheds according to 3 dominant use types. We compared relative areas (land covers as dependent variables) classifying four categories (native forest, mature forest plantation, inmature forest plantation and agricultural activities for the entire watershed and riparian areas.

L37: Unclear what is meant by ‘DOM metrics’

Response 7:  dom characteristics

Methods

L177: Why would you perform a Shapiro-Wilk test for land cover data and DOM metrics? You are hopefully aware that the assumption of an ANOVA and a linear model are the normality, which means the normal distribution of theresiduals, and the homogeneity of variance.

Response 8: we improve the wording of this section

L184: What are the DOM quantity and quality parameters? Be consistent in the terminology used! DOM metrics, DOM characteristics, DOM parameter. Are these all the same? And what are the qualitative ones? What are ‘the land cover data’ - provide the variables used in the analysis. And what are the ‘scales of watersheds’?

Response 9:  DOM characteristic according to quantity and quality

L185: Why the z-standardization?! You use rank based approaches. Z-Standardization doesn’t make sense here.

Response 10: ok

Figure 1: How are the three maps with A to L related to each other? Provide a map of the basin where map sections are related to each other. It looks like a very nested design, where all native forest areas are closer to each other than to the other forest types.

Response 11: we will provide a map of the basin where map sections are related to each other.

L192: Specify the two independent variables. ‘Land cover of native forests to scales of analysis riparian area at 60 and 30 m.’ This is not a proper English sentence, and it is unclear what exactly the (2, 3 or more?) independent variables are.

Response 12: We used univariate regression analyses to test for the relationship between DOM quantity and quality (DIC and C2 component) and native forest cover in riparian area at 60 and 30 meter wide buffers (independent variables).

Methods are unclear. As far as I understand you use ANOVA plus PostHoc test with the Null-Hypothesis: ‘There are no significant differences in the DOM between the four land cover types: native forest, young plantation, old plantation, agriculture in the watersheds A to L’. But apparently DOM is measured by several indicators, so you have several ANOVAs. How many? Consider multiple testing!

Response 13: Non-parametric statistical analyses such as the Kruskal Wallis and Nemenyi tests of multiple comparisons for independent samples (tukey) were used to determine significant differences among watersheds dominated by forest plantations and agricultural lands with reference.

The second step is the Spearman correlation of DOM parameters (still unclear of how many and which ones they are) and the land cover data. What variables were correlated? Consider again multiple testing! Why (according to Figure S3) the Spearman correlation was performed only for the forest plantation and agriculture watershed?

Response 14: we selected only dependent variables that presented significant differences with a p value of <0.05 (Table S2) according to Spearman rank correlations. These matrixes were created to relate DOM quantity and quality and land cover in the entire watershed and riparian area (30 and 60 m wide buffers).

The third step are several linear models, as far as I understand with the same response variables than in the ANOVA, only that there are several more independent variables. Then the linear models are redundant to the ANOVA and it is unclear why to use both.

Response 15: we improve the wording of this section

Results

L205: Replace 0,05 by 0.05 and be consistent in the use of a digit separator.

Response 16: ok

L207: Freshness index. It is the first time you talk about such thing and it is completely unclear where it comes from, or whether it is any synonym to something you have talked about before. If this is something measured or calculated it needs to be mentioned in the methods.

Response 17: the freshness index was calculated and explain in methods

Conclusion

L319 New insights – is unspecific. Please specify what are these insights.

Response 18: we improve the wording of this section

Reviewer 2 Report

I have completed my review of “The role of native riparian forests on dissolved organic matter in forest plantation and agriculture watersheds in northwestern Patagonia” by Becerra-Rodas et al. This is a very enjoyable paper and has been placed into the correct section of this journal. The results and discussion could use a bit more work specifically in clarifying a few points and expanding on some of the findings. However, this is purely an extension of the work already done and does not require any more statistical analysis so should not be a difficult undertaking for the authors. I have also noted several small edits throughout the text which area easily corrected. Tables and Figures have been used well in this manuscript and the inclusion of data in Supplementary Information was well done with the appropriate content being placed in this section. I have selected Minor Revisions for this manuscript as I feel with a little bit of work on the results and discussion this paper can be improved for clarity to the reader.

Line 46 – write out the full taxonomic name for all species, in this case for Pinus radiata, on the first occasion within the text. A good resource to find this is at http://www.theplantlist.org/

Lines 49-50 – “associated with the maintenance”

Line 53 – “native riparian forests”

Line 65 – “this is the most common metrics of DOM” – two tenses are used here, please select the most appropriate one

Line 78 – “suggest that forest ecosystem disturbance, such as clear cutting, induce changes”

Lines 80-82 – This sentence does not read well. I would recommend restructuring it.

Line 90 – “northwest Patagonian watersheds”

Line 93 – “cover” singular

Line 95 – “watershed” singular

Line 99 and Line 267 – capitalize “Valdivian”

Line 102 – change to either “forest cover will dominate humic-like” or “forest cover will be dominated by humic-like”

Line 103 – “inputs structurally complex” - it is unclear what you mean by this (structural complexity?) - please reword

 Line 112 – “sawn wood industry” is not a term that is typically used. It is not incorrect but I would suggest the authors consider changing this to “lumber industry”

Line 120 – “that has accessibility through roads as well as similar topography”

Line 128-9 – This sentence needs a rewrite. I am unsure if you interpreted the data or if you used a freely available interpretation.

Line 129 – remove “authors”

Line 134 – “analyst” ? or did you mean “analysis”

Line 144 – “during various levels of direct runoff” ? I do not understand what you mean by this.

Line 145 – “connected to obtain”

Line 154 – within 2 days of what? This is unclear – is this 2 days from sampling or 2 days from returning the samples to the lab or were they stored at 7⁰C for 2 days specifically?

Line 159 – “units were generated”

Lines 168 and 169 – I think you want to use a period (.) not a comma (,) when you use 1.9 and 1.4

Line 169 – terrestrially derived what? please clarify

Lines 169-171 – your description of freshness index values needs to be clarified. You state values under 1 mean this is recently formed but less than 0.6 means ancient formed DOM. This cannot be as then if a value was say 0.4 according to your statement it is both recently formed and ancient formed. Did you mean to indicate that from 0.6 to 1.0 is recently formed or should it be greater than 1 is recently formed?

Line 177 – rewrite please – did Sharpiro-Wilk undertake a test or did you perform a Shapiro-Wilk test? This is not clear from this sentence.

Lines 190-192 – sentence needs to be rewritten for clarity please

Lines 196-199 and Figure 3 – These are some really great findings about the different compositions of land cover in the different watersheds. Yet very little is said about them (4 lines in Results and nothing directly linked to this in the Discussion). I think it would be worthwhile going into a bit more detail around these findings. Looking at Figure 3 there are major changes between the different plots and the composition of the riparian areas for the different types of watersheds is also variable. Does this have any effect on your findings? For example the agricultural watersheds vary widely in the coverage between the entire watershed and the riparian zones – are differences in your other findings related to this or did you just look at the watershed groups as a whole. I would like to see a bit more discussion around the results seen in this section.

Lines 206-207 – were all the concentrations statistically significant? please clarify in the text if it was all or some or just the average values. I know you link this to Table S2 which is good but as this is a supplementary info table you need to be very clear about what you include in the text as many readers won’t check you supplementary info (as it should serve only as reference information for those wishing greater knowledge of your study).

Lines 227-233 and Figure 4 – why have concentration of DIC for forest plantation watersheds and C2 Fmax for agricultural lands been selected for representation in Figure 4? Why are the other metrics not displayed? Was this because these were the most important variables? Please expand this paragraph to indicate why this has occurred and why the other metrics are not shown. Figure 4 is a really good way of displaying this data so it is just in the text that clarification is needed.

Lines 246-247 – is this a finding from this study or a general statement based on the literature. It would be good to know that as it is unclear if this is a finding or a known fact that is the cause for the assessment of the findings.

Line 257-258 – how to fertilizers, herbicides, and cattle manure related to DIC and DOM. Don’t assume your reader will know the link – you need to clarify this for them.

Line 260 – “tributaries of the Mississippi river”

Line 267 – “rainforest” one word

Line 268 – Is the Nimptsch et al control sample site also in the Valdivian temperate rainforest? The other examples you give are not (which is fine) but it is not clear what type of forest the Nimptsch samples are found in.

Line 275 – change “references” to “reference sites” or “reference locations” or “reference watersheds”

Line 289 – “degree of humification”

Lines 308-309 – please rewrite as it currently reads as if allochthonous DOM would cause increases in native forest cover in the riparian areas.

Discussion – I would like to see something added in about the differences noted between your results from the 60m and 30m riparian zone buffers. The conclusions note that “Results of this study are useful to inform policy and regulations about the width of native forest buffers as well as their characteristics and restrictions” (lines 328-329). However, without comparing and contrasting the two widths you have assessed it is difficult to determine what is an optimal width of native forest in different land covered reaches based on your results. I am certain that your results provide this information but you must cover it in your discussion to prove this conclusion (and I think this is something worthy of discussing).

Table 1 – I really like how you have included this information in a table in this manner – it is a clear and easy way to find all this data. Just a few notes on the layout of the table. Currently the Headings don’t fit well in the width of the columns you have provided so I would recommend that be modified. You could also place the table on its own page and make it landscape orientation which would allow for wider columns. There is also a minor issue with continuity – you use a line delineating the “land cover description in watershed” between rows A and B but not between rows F and G, G and H, and K and L where it also appears that this type of delineation should be made. Also a minor edit to the caption is suggested as “number of sample locations, and description of land cover”

Table 2 – you only need to include the legend on what T, A, and M mean once so you can either have it in the caption or as a line at the bottom of the table – either is correct but you don’t need both.

Table 2 – I believe in the Emission max (nm) column you should be using a period (.) not a comma (,)

Table 3 – I think this would also be best served if presented in landscape form as the data is quite bunched up when presented as a portrait orientated table.

Figure 1 – I really like this figure, while there is a lot displayed on the images the context of all items is clear which is important. However, it is hard to see the site numbers. Perhaps a larger font would be good and movement of some numbers (eg. 2, 5) so that the dots are not directly on top of them.

Figure 2 – I like that you have included an example photo here – it makes your methods clear. However, I cannot tell what your buffer zones are as you have used red lines on a green image (which is an issue for those who are colour blind). Please change the red colour to either a colour that will stand out better on the green (bright yellow perhaps) or use another delineation method such as cross hatching in black as this would mean that the reader could tell what the areas are even if they are looking at your paper in black and white.

Figure 3 – I like the layout of this figure but again the colours are very difficult to delineate from one another. Please select colours that are less similar or you can present this figure in black and grey using cross hatching and other patterns for the different land use types. I would also suggest perhaps making the bars a bit wider with less white space between them which will help make it easier to see where the changes are.

Figure S3 – I really like this figure – it is a unique way to present a large amount of data while still being very clear on what is happening for the data series. The shortforms that you use in the Figure are never explained though and I would recommend adding in either a legend or information into the caption. Remember this is the supplemental info so anyone who is looking through this wants greater information on your study and will be happy to sort through this type of information (rather than a main text figure where this could be overly complicated. Also it would be good to have a better range on what the size of the circles means. I see that a circle means statistical significance and I am assuming that a larger circle means a larger significance but a scale showing this range would be a good inclusion to this figure. Also why are there fewer rows and columns in the Agriculture lands figure? Please clarify.

Author Response

Response to Reviewer 2 Comments

Line 46 – write out the full taxonomic name for all species, in this case for Pinus radiata, on the first occasion within the text. A good resource to find this is at http://www.theplantlist.org/

Response 1:  we changed for Pinus radiata D.Don. and Eucalyptus globulus Labill

Lines 49-50 – “associated with the maintenance”

Response 2:   ok

Line 53 – “native riparian forests”

Response 3:   ok

Line 65 – “this is the most common metrics of DOM” – two tenses are used here, please select the most appropriate one

Response 4:   ok

Line 78 – “suggest that forest ecosystem disturbance, such as clear cutting, induce changes”

Response 5:   ok

Lines 80-82 – This sentence does not read well. I would recommend restructuring it.

Response 6:   ok

Line 90 – “northwest Patagonian watersheds”

Response 7:   ok

Line 93 – “cover” singular

Response 8:   ok

Line 95 – “watershed” singular

Response 9:   ok

Line 99 and Line 267 – capitalize “Valdivian”

Response 10:   ok

Line 102 – change to either “forest cover will dominate humic-like” or “forest cover will be dominated by humic-like”

Response 11:   ok

 Line 103 – “inputs structurally complex” - it is unclear what you mean by this (structural complexity?) - please reword

Response 12:    Specifically, we hypothesize that the increase in native riparian forest cover will be dominated by humic-like DOM components inputs to streams, with structural complexity and more transport downstream

 Line 112 – “sawn wood industry” is not a term that is typically used. It is not incorrect but I would suggest the authors consider changing this to “lumber industry”

Response 13:     “lumber industry”

Line 120 – “that has accessibility through roads as well as similar topography”

Response 14:    ok

Line 128-9 – This sentence needs a rewrite. I am unsure if you interpreted the data or if you used a freely available interpretation.

Response 15:    we improve the writing

Line 129 – remove “authors”

Response 16: Ok

Line 134 – “analyst” ? or did you mean “analysis”

Response 17: Ok

Line 144 – “during various levels of direct runoff” ? I do not understand what you mean by this.

Response 18:  Water samples were collected during the winter season in July and August, especially direct runoff because soils were more saturated and flowpaths were connected for obtain terrestrial signal

Line 145 – “connected to obtain”

Response 19: Ok

Line 154 – within 2 days of what? This is unclear – is this 2 days from sampling or 2 days from returning the samples to the lab or were they stored at 7⁰C for 2 days specifically?

Response 20: Ok

Line 159 – “units were generated”

Response 21: Ok

Lines 168 and 169 – I think you want to use a period (.) not a comma (,) when you use 1.9 and 1.4

Response 22: Ok

Line 169 – terrestrially derived what? please clarify

Response 23: Ok

Lines 169-171 – your description of freshness index values needs to be clarified. You state values under 1 mean this is recently formed but less than 0.6 means ancient formed DOM. This cannot be as then if a value was say 0.4 according to your statement it is both recently formed and ancient formed. Did you mean to indicate that from 0.6 to 1.0 is recently formed or should it be greater than 1 is recently formed?

Response 24: Ok

Line 177 – rewrite please – did Sharpiro-Wilk undertake a test or did you perform a Shapiro-Wilk test? This is not clear from this sentence.

Response 25: we improve the wording of this section

Lines 190-192 – sentence needs to be rewritten for clarity please

Response 26: Ok

 Lines 196-199 and Figure 3 – These are some really great findings about the different compositions of land cover in the different watersheds. Yet very little is said about them (4 lines in Results and nothing directly linked to this in the Discussion). I think it would be worthwhile going into a bit more detail around these findings. Looking at Figure 3 there are major changes between the different plots and the composition of the riparian areas for the different types of watersheds is also variable. Does this have any effect on your findings? For example the agricultural watersheds vary widely in the coverage between the entire watershed and the riparian zones – are differences in your other findings related to this or did you just look at the watershed groups as a whole. I would like to see a bit more discussion around the results seen in this section.

Response 27: we improve this figure and its description in the written

Lines 206-207 – were all the concentrations statistically significant? please clarify in the text if it was all or some or just the average values. I know you link this to Table S2 which is good but as this is a supplementary info table you need to be very clear about what you include in the text as many readers won’t check you supplementary info (as it should serve only as reference information for those wishing greater knowledge of your study).

Response 28: add significance code in table 3

Lines 227-233 and Figure 4 – why have concentration of DIC for forest plantation watersheds and C2 Fmax for agricultural lands been selected for representation in Figure 4? Why are the other metrics not displayed? Was this because these were the most important variables? Please expand this paragraph to indicate why this has occurred and why the other metrics are not shown. Figure 4 is a really good way of displaying this data so it is just in the text that clarification is needed.

Response 28: we improve the description in the written

Lines 246-247 – is this a finding from this study or a general statement based on the literature. It would be good to know that as it is unclear if this is a finding or a known fact that is the cause for the assessment of the findings.

Response 29:  Remove word effectiveness

Line 257-258 – how to fertilizers, herbicides, and cattle manure related to DIC and DOM. Don’t assume your reader will know the link – you need to clarify this for them.

Response 30:  we improve the wording of this lines

Line 260 – “tributaries of the Mississippi river”

Response 31:  ok

Line 267 – “rainforest” one word

Response 32:  ok

Line 268 – Is the Nimptsch et al control sample site also in the Valdivian temperate rainforest? The other examples you give are not (which is fine) but it is not clear what type of forest the Nimptsch samples are found in.

Response 33:  Molco river associated to Nothofagus-dominated temperate rainforests

Line 275 – change “references” to “reference sites” or “reference locations” or “reference watersheds”

Response 34:  ok

Line 289 – “degree of humification”

Response 35:  ok

Lines 308-309 today line 346– please rewrite as it currently reads as if allochthonous DOM would cause increases in native forest cover in the riparian areas.

Response 36:  ok.

Discussion – I would like to see something added in about the differences noted between your results from the 60m and 30m riparian zone buffers. The conclusions note that “Results of this study are useful to inform policy and regulations about the width of native forest buffers as well as their characteristics and restrictions” (lines 328-329). However, without comparing and contrasting the two widths you have assessed it is difficult to determine what is an optimal width of native forest in different land covered reaches based on your results. I am certain that your results provide this information but you must cover it in your discussion to prove this conclusion (and I think this is something worthy of discussing).

Response 37:  The characteristics of DOM within watersheds dominated by forest plantations and agriculture lands might be partially attributed to the gradient of native riparian forest coverage in the riparian area, principally buffers of 60 meters wide, which is more than required by Chilean law for headwater streams

Table 1 – I really like how you have included this information in a table in this manner – it is a clear and easy way to find all this data. Just a few notes on the layout of the table. Currently the Headings don’t fit well in the width of the columns you have provided so I would recommend that be modified. You could also place the table on its own page and make it landscape orientation which would allow for wider columns. There is also a minor issue with continuity – you use a line delineating the “land cover description in watershed” between rows A and B but not between rows F and G, G and H, and K and L where it also appears that this type of delineation should be made. Also a minor edit to the caption is suggested as “number of sample locations, and description of land cover”

Response 38:  we corrected this table

Table 2 – you only need to include the legend on what T, A, and M mean once so you can either have it in the caption or as a line at the bottom of the table – either is correct but you don’t need both.

Table 2 – I believe in the Emission max (nm) column you should be using a period (.) not a comma (,)

Response 39:  we corrected this table

Table 3 – I think this would also be best served if presented in landscape form as the data is quite bunched up when presented as a portrait orientated table.

Response 40: ok 

Figure 1 – I really like this figure, while there is a lot displayed on the images the context of all items is clear which is important. However, it is hard to see the site numbers. Perhaps a larger font would be good and movement of some numbers (eg. 2, 5) so that the dots are not directly on top of them.

Response 41: ok 

Figure 2 – I like that you have included an example photo here – it makes your methods clear. However, I cannot tell what your buffer zones are as you have used red lines on a green image (which is an issue for those who are colour blind). Please change the red colour to either a colour that will stand out better on the green (bright yellow perhaps) or use another delineation method such as cross hatching in black as this would mean that the reader could tell what the areas are even if they are looking at your paper in black and white.

Response 42: ok 

Figure 3 – I like the layout of this figure but again the colours are very difficult to delineate from one another. Please select colours that are less similar or you can present this figure in black and grey using cross hatching and other patterns for the different land use types. I would also suggest perhaps making the bars a bit wider with less white space between them which will help make it easier to see where the changes are.

Response 43: ok 

Figure S3 – I really like this figure – it is a unique way to present a large amount of data while still being very clear on what is happening for the data series. The shortforms that you use in the Figure are never explained though and I would recommend adding in either a legend or information into the caption. Remember this is the supplemental info so anyone who is looking through this wants greater information on your study and will be happy to sort through this type of information (rather than a main text figure where this could be overly complicated. Also it would be good to have a better range on what the size of the circles means. I see that a circle means statistical significance and I am assuming that a larger circle means a larger significance but a scale showing this range would be a good inclusion to this figure. Also why are there fewer rows and columns in the Agriculture lands figure? Please clarify.

Response 44: we selected only dependent variables that presented significant differences with watershed of reference with a p value of <0.05 (Table S2)

Round 2

Reviewer 1 Report

Dear authors,

You did not answer to my comments as expected. Mainly in the method section, half of the questions and points I have raised, you did not even reply to, e.g. multiple testing, why z-standardization was initially chosen and what exactly are the independent variables. This raises serious concerns whether the statistical analysis is properly done and as long as the methods are not described well enough to follow the individual steps, it is not possible to evaluate the reliability of the outcome.

Author Response

Response to Reviewer 2 Comments

 Point 1: You did not answer to my comments as expected. Mainly in the method section, half of the questions and points I have raised, you did not even reply to, e.g. multiple testing, why z-standardization was initially chosen and what exactly are the independent variables. This raises serious concerns whether the statistical analysis is properly done and as long as the methods are not described well enough to follow the individual steps, it is not possible to evaluate the reliability of the outcome.

Response 1:

 We modificated this section by this .."In order to asses significant differences between different conditions compared with reference, we used non-parametric analysis of variance, Kruskal Wallis (ggpubr R packages [53]) and Nemenyi tests of multiple comparisons for independent samples (tukey) (PMCMR R package [54, 55]). This analysis identified a subset de DOM parameters which differed significantly by watershed type (p<0.05, Table 3 and Table S2). DIC concentrations and intensities of the maximum fluorescence of PARAFAC components were related to land cover portion in eight watersheds (WFP and WAL) using Spearman rank correlations (cor.test function in R corrplot R package [56]). Land cover included percent cover of native forest (NF), mature forest plantation (MFP), young forest plantation (YFP) and agricultural land (AL) as a proportion of the entire watershed and buffer strip of 30 and 60 m width. Finally, the two DOM parameters that differed significantly for watershed dominated by forest plantation and agricultural land (DIC concentrations and C1 and C2 components as humic, C3 and C4 components as protein, were related to streamside native forest in buffer strip of 30 and 60 meter width using linear model (lm function in R). Statistics were performed using R version 3.2 [57], Rstudio [58], ggplot, R package designed to create and customize plots [59]."
